# Foot Placement in the Basic Position on the Start Block OSB12 of Young Competitive Swimmers

**DOI:** 10.3390/sports12020042

**Published:** 2024-01-29

**Authors:** Ivan Matúš, Bibiana Vadašová, Tomáš Eliaš, Łukasz Rydzik, Tadeusz Ambroży, Wojciech Czarny

**Affiliations:** 1Department of Sports Kinanthropology, Faculty of Sports, Universtiy of Presov, 08001 Prešov, Slovakia; ivan.matus@unipo.sk (I.M.); bibiana.vadasova@unipo.sk (B.V.); tomas.elias@unipol.sk (T.E.); wojciechczarny@wp.pl (W.C.); 2Institute of Sports Sciences, University of Physical Education, 31-571 Kraków, Poland; tadek@ambrozy.pl; 3Institute of Physical Culture Studies, College of Medical Sciences, University of Rzeszow, 35-959 Rzeszów, Poland

**Keywords:** kick start, kinematic analysis, biomechanics, start performance

## Abstract

Background: The basic position on the starting block can influence the performance at the start, as it is the initial phase on which the other phases depend, as well as the swimming performance in sprint events in all swimming styles. The aim of our study is to analyze the effect of the foot in the base position on the block start on performance in the 5 m distance start. Material and Methods: Fifteen performance swimmers aged 17 ± 2 years were tested in their preferred wide and narrow starting positions, performing a total of six starts during which angular, temporal, and length changes were monitored in block, flight, and underwater phases. Fisher individual tests for differences of means were used to determine differences in kinematic parameters of the kick start to the 5 m distance. Differences in the position of the feet in kinematic parameters of the kick start to the 5 m distance were determined using the two-sample *t*-test with equal variance and effect size by Cohen’s d. Results: Swimmers were found to have significant differences (*p* < 0.05) between foot widths in block time (0.02 s), time to 2 m (0.05 s), flight and glide time and distance, maximal depth, and time to 5 m (0.08) in favor of the narrow baseline position. Conclusions: We recommend marking the center of the start block on the OSB or OSB platform for the competitors, as well as the center of the backrest, for better orientation and assuming the correct basic foot position on the start block.

## 1. Introduction

The swimmer’s performance in the sprint events is getting shorter and shorter. This is certainly due to the starting block of OSB11 or its platform OSB12. A new facelift of the OSB11 starting block has been introduced for the Budapest 2022 World Swimming Championships, and swimmers have not yet started from it. The performance at the start is conditioned by the individual phases that follow each other. From the first block phase, the swimmer transitions to the flight phase, from the flight phase to the underwater phase, and free swimming to 15 m. At the 15 m line, the swimmer’s head must break the surface of the water, and the swimmer must start swimming in the stroke (except breaststroke, FINA, 2023). Improvements in launch performance have been addressed by several authors [1,2,3,4,5,6] who have demonstrated the significance of differences between different levels (1–5) of the rear kick plate. Other studies looked at the change in baseline body position, which was assessed based on shoulder position or center of gravity position, depending on the material equipment in the studies. Three positions were monitored—front, neutral, and back. In most studies, shorter times were measured at 5 m, 7.5 m, and 15 m, respectively, at neutral-weighted or rear-weighted positions [6,7,8,9,10]. In the study by Burkhardt et al. [11], the positioning of the lower limbs in the base position on the starting block did not show differences in the base position used between the stronger lower limb on the front edge or on the backrest. Nevertheless, they did find some differences in the strength (higher horizontal force and impulse) of the lower limbs when jumping from the backrest, so they recommended placing the stronger lower limb on it. In the study by Matúš et al. [9], they did not focus on the strength of the lower limbs but on their dominance and position in the base position on the start block. The study showed that swimmers achieved a shorter 5 m distance time if the dominant lower limb was located at the front edge of the starting block OSB12. Similar findings were obtained in the study by Takeda et al. [4], where the lower limb at the front edge mainly generated vertical take-off velocity and the rear lower limb generated horizontal take-off velocity. In a study by Silveira et al. [12], they found that after the kick-off from the start block, kick start and its modification showed that the key parameter was the flight length at the 5 m take-off distance. In a study by Matúš et al. [8], start reaction, take-off angle, and time to 2 m distance were shown to be the key parameters in the 5 m distance start. When looking for key parameters at a 15 m distance in the study by Tor et al. [13], length and time of flight were not among the key parameters affecting take-off performance. Horizontal take-off speed (81%) was found to be an important parameter in the flight phase, with the parameters divided into above and underwater parameters. Backstroke stance was addressed in a study by Slawson et al. [14], which demonstrated significant differences between elite swimmers (both male and female) on an output measurement due to a change in base stance width in favor of a narrow base stance compared to a wide base stance. In a study by Kibele et al. [15], while following elite swimmers, they analyzed block time, start times to 5 m, horizontal take-off velocities, and horizontal vs. vertical peak force values at the start of the OSB11. Several changes were included in the study, such as the exchange of the front lower limb (left vs. right), the center of mass (CM) height (low vs. high), the stance width (narrow vs. wide), and rear vs. front weight stance. Results indicated a forward and higher CoM position on the block with a narrow stance of the back plate and a forward and lower CoM position on the block with a wide stance of the back plate showed the highest advantages in block time, horizontal peak force, and time to 5 m.

Most of the studies dealt with elite swimmers, where the swimmers received qualitative feedback, which helped them to optimize their baseline position on the block start and, thus, improve their overall start performance. On the other hand, there is a lack of studies dealing with performance swimmers, of which there are certainly more, who could improve their performance in sprint events in swimming based on such an analysis. The aim of our study is to analyze the effect of the foot in the base position on the block start on performance in the 5 m distance start. The information obtained may be useful for optimal adjustment of the basic position on the block start for more young competitive swimmers.

## 2. Materials and Methods

### 2.1. Participants

Fifteen competitive swimmers aged 17 ± 2 years, with mean body height 186 ± 0.07 cm and mean body weight 79.9 ± 5.0 kg, participated in the measurement. The swimmers regularly participate in Slovak championships. Each of them focuses on sprinter disciplines in swimming. The average FINA score in the 50 m freestyle short-distance pool in the 1 month prior to the study was 630 points. Before the measurement, each of the swimmers confirmed that they had no health problems. Before the tests began, the swimmers were informed about the procedure and gave their written consent. Each of the swimmers was instructed that they could withdraw from testing at any time. Ethical approval for this study was obtained from the Ethics Committee of the University of Presov, Presov, Slovakia (Approval No. ECUP042022PO).

### 2.2. Test Protocol

The research was carried out at the 25 m indoor pool of the University of Presov, which has 6 lanes (water depth 1.56–1.86 m), which is not equipped with new OSB starting blocks, so we used the OSB12 starting platform in the measurement. In accordance with the methodology [8,9,16], points were marked on the swimmers’ bodies with a black waterproof marker. The application of the 11 dots was performed under the supervision of a doctor from the Faculty of Sport of PU as follows: lateral margin of the left transverse tarsal joint; lateral left and right malleolus; lateral left and right knee condyle; left and right greater trochanter; lateral margin of the left and right scapular spine; lateral left and right elbow epicondyle; ulnar styloid process of the left and right wrist, and medial side of the 5th metacarpal–phalanx joint.

Before the measurement, the swimmers did a standard 10 min warm-up according to the RAMP protocol:Raise—increase muscle temperature, core temperature, blood flow, muscle elasticity, and neural activation;Activate—engage the muscles in preparation for the upcoming session;Mobilize—focus on movement patterns which will be used during the activity;Potentiate—gradually increasing the stress on the body in preparation for the upcoming competition/session.

This was followed by a 400 m swim under the supervision of the coach. The swim was followed by three test jumps according to the preferred position. After that, an instruction on the basic position of the block start was carried out. The basic start position on the start block was the preferred basic position on the start block from which the feet on the starting block were in a narrow or wide stance. The narrow position of the feet in the starting position was one foot placed behind the other (±5 cm from the center of the start block). From this stance, the wide stance was based on the feet being set one foot to the side (shoulder-width apart).

The starting commands were executed in accordance with FINA rules. Swimmers were instructed not to perform any kicking or undulating movements during the underwater phase. According to the study by Tor et al. [13], swimmers should only perform the first kick at 6.6 m; therefore, swimmers only swam within 5 m of the pool wall. The swimmers performed 3 kick starts at a 5 m distance in a narrow position and 3 kick starts at a 5 m distance in a wide position. In total, the swimmers performed 6 kick starts. After three kick starts, there was a break of 30 min. The order of kick starts (narrow or wide) was random. Testing was done in a continuous form, where after the first one started, the next one started after 30 s. This meant that the rest between kick starts for each swimmer was sufficient −7.30 min. In the case of an incorrect base position, the swimmer was instructed to assume the correct base foot position during the base position at the start block. The duration of the load in a single start was mainly on the start block, and it lasted approximately 0.5–0.7 s.

To perform the kick starts, an OSB12 start platform was used and attached to the start block (tape went across the middle to divide the start platform in two), as the pool did not have new OSB11 start blocks. The OSB12 starting block has these dimensions: 740 mm × 520 mm × 38 mm (165 mm with footrest). The angle of the starting block was 9°. The backrest has an adj. range of 200 mm (in 5 steps). The angle of the backrest was 30° (Figure 1). The height of the starting block from the water surface was 700 mm.

The Swimpro camera system was used to analyze the kick starts. The location of the cameras can be seen in the figure (Figure 2). The cameras were placed at a distance of −1 camera 0 m from the edge of the pool at a camera height of 1.5 m. The second and third cameras were at the same distance from the edge of the pool, 1.6 m, with the camera above the water at a height of 1.5 m and the camera underwater at a depth of −1.7 m. The fourth camera was at 5 m from the edge of the pool at a depth of −1.7 m. The camera system was operating at 50 fps at a shutter speed of 1/1000 s. The pool area was illuminated with additional halogen light above the cameras. Additional supplementary LED lighting was placed along the edge of the pool.

Using the Dartfish© software (Dartfish ProSuite 4.0, 2005; Fribourg, Switzerland), the 2D analysis of video recordings was performed to evaluate the following phases and kick start parameters [17,18]:*block phase*—FKA front knee angle (°), FAA front ankle angle (°), RKA rear knee angle (°), RAA rear ankle angle (°), HA hip angle (°), BT block time (s);*flight phase*—TA take-off angle (°), T2 time to 2 m (s), T2 velocity to 2 m (m/s), EA entry angle (°), FT flight time (s), FD flight distance (m);*underwater phase*—GT glide time (s), GD glide distance (m), MaxH maximal depth (m), T5 time to 5 m (s), T5 velocity to 5 m (m/s).

Fisher individual tests for differences of means were used to determine differences in kinematic parameters of the kick start to the 5 m distance. Differences in the position of the feet in kinematic parameters of the kick start to the 5 m distance were determined using the two-sample *t*-test with equal variance. Significant differences were assessed at *p* < 0.05. Effect size based on mean comparison was determined by Cohen’s d (small effect = 0.2, medium effect = 0.5, and large effect = 0.8). The statistical software used was Statistica 14.

## 3. Results

### 3.1. Block Phase

The descriptive statistics show that the mean block time (resulting time for block phase) for the narrow basic position was 0.77 ± 0.03 s. The mean block time for the wide basic position was 0.79 ± 0.02 s (Figure 1). The one-sided hypothesis testing demonstrated that at the α = 0.05 level of significance, the difference in the block time between narrow and wide basic positions was *p* = 0.001. The two-sample *t*-test showed that block times were significantly shorter for the narrow position than the wide basic position. The effect size for the block time (d = −0.72) was found to exceed Cohen’s (1988) convention for a medium effect (Table 1). This difference was achieved for the same lower limb angles (FAA, FKA, RKA, RAA), as well as for trunk angle, where we did not find statistical significance of the differences.

### 3.2. Flight Phase

The descriptive statistics show that the average time to 2 m for the narrow basic position was 0.94 ± 0.02 s. The mean time to 2 m for the wide basic position was 0.99 ± 0.02 s. The same was true for the speed conversion. The flight time to the narrow basic position was 0.37 ± 0.01 s, and for the wide basic position, it was 0.41 ± 0.01 s. The flight distance to the narrow basic position was 2.69 ± 0.07 m, and for the wide basic position, it was 2.61 ± 0.07 m. The one-sided hypothesis testing demonstrated that at the α = 0.05 level of significance, the difference in the time to 2 m and flight time and flight distance between the narrow and wide basic positions was *p* = 0.001. The two-sample *t*-test showed that time to 2 m and flight time were significantly shorter time, with longer flight distance for the narrow position than the wide basic position. The effect size for the time to 2 m (d = −2.11; 2.16), flight time (d = −3.40), and flight distance (d = 1.12) were found to exceed Cohen’s (1988) convention for a large effect (Table 2).

### 3.3. Underwater Phase

The descriptive statistics show that the average time to 5 m for the narrow basic position was 1.67 ± 0.03 s. The mean time to 5 m for the wide basic position was 1.75 ± 0.03 s. The same was true for the speed conversion. The glide time to the narrow basic position was 0.52 ± 0.02 s, and for the wide basic position, it was 0.55 ± 0.02 s. The glide distance to the narrow basic position was 2.31 ± 0.07 m, and for the wide basic position, it was 2.39 ± 0.07 m. The maximal depth to the narrow basic position was −0.89 ± 0.02 m, and for the wide basic position, it was −0.92 ± 0.07 m. The one-sided hypothesis testing demonstrated that at the α = 0.05 level of significance, the difference in the time to 5 m and glide time, glide distance, and maximal depth between the narrow and wide basic positions was *p* = 0.001. The two-sample *t*-test showed that time to 5 m and glide time were significantly shorter time, with shorter glide distance and maximal depth for the narrow position than the wide basic position. The effect size for the time to 2 m (d = −3.41; 3.40), glide time (d = −2.11), glide distance (d = −1.13), and maximal depth (d = 1.4) were found to exceed Cohen’s (1988) convention for a large effect (Table 3).

## 4. Discussion

The aim of the study was to analyze the effect of foot position in basic stance on the start block on the performance in the 5 m distance start in performance swimmers. To investigate the effect of changing the base stance on the start performance, the swimmers could not perform waves or kicking underwater; they could only swim.

In competitive swimming events, start, swim distance, turns, and finish are extensively analyzed [19]. However, without the start, the race could not take place because the opponents could not start equally. As this is the initial phase, it needs to be given sufficient attention, especially in sprint disciplines, as it is followed by other phases. The advantages provided by OSB starting blocks or their platforms lie in the rear support and its adjustment, which allows displacement in five positions. This change in the start block has led to some research [2,4,6,7,12,13,20], which have addressed the adjustment of the backrest or the change of the base position on the start block, which affect the kinematic–dynamic parameters in the on-start block phase, the kinematic parameters in the flight and underwater phases, as well as the take-off performance itself. For example, a study by Cicenia et al. [21] looked at the use of shin length as a measure to determine cleat position and its effect on performance. The results of the study indicated that shin length is a quick and individualized measure that coaches can use to determine kicking plate position without compromising performance. A study by Shepherd et al. [7] demonstrated that rebounding body position does not differ for the two genders but is a critical factor in determining initial performance for both genders. Barlow et al. [1] and Matúš and Kandráč [5] reported the best performance when the kick plate is at the third and fourth level, respectively. Starting performance is also influenced, to some extent, by experience gained during previous practice [22], and regular training can improve the engagement of motor units [23] that can influence starting performance, especially during take-off from the start block, where the swimmer can reach twice the speed as opposed to swimming [13,24]. The benefits of a preferred base position on the start block have also been demonstrated in a more recent study by Rudnik et al. [6] in both male and female swimmers in a 15 m start. The better performance achieved in the preferred base position may also be due to a psychological effect or the result of prolonged motor learning in the kick starts. On the other hand, a preferred backrest setup on an OSB or OSB platform may lead to a shorter take-off performance, but the basic position on the start block before the take-off is still crucial [5,8,9]. Therefore, we can distinguish three basic positions in the base position on the start block-front-, neutral-, and rear-weighted [1,15,25,26], with neutral-weighted and weighted being confirmed in some studies [1,5,9]. A change in baseline position can directly affect performance on the start block (block time), either positively or negatively [2,4,24,27], or there may be no difference at all [6,28].

Some studies have shown that joints, such as the knee, can influence force transmission and, therefore, flight performance [29]. The lower limbs in the base position contribute to differential velocity production when taking off from the starting block [30,31]. In addition to the changes that affect the start in swimming, some studies have also looked at the width of the feet in the base position on the start block [14,15,32], which came to the findings that a narrow base position shortened time on the start block and generated higher horizontal maximal force with a high body center of gravity. All three studies followed elite swimmers, where groups were made up of both men and women, and in some of the studies, the start blocks were not OSB but replicas of them. The results of our study showed the effect of varying foot width in the preferred base position on the start block, which was mainly observed after the take-off from the OSB12 and in the kinematic parameters in the flight and underwater phases. Similarly, the performance at a 5 m distance showed significant differences between stance width and exit or performance parameters. Similar results were obtained in the study by Slawson et al. [14], where foot width in the baseline position influenced the variables studied as well as the performance itself over a 15 m distance of elite male and female athletes. In this study, they also looked at a case study of an individual athlete, stating that such a study creates better insight for effectively assuming a baseline position on the start block for the swimmer being measured. Fischer [33] reported that wide stride lengths were associated with a deferred force development in both legs after the take-off from the OSB9 (replica).In a study by Kibele et al. [15], in most swim starts, improvements were associated with a narrow stance and increased body center of gravity. Shorter times in block time were observed in such a baseline position. A shorter block time was also demonstrated in our study in the narrow foot base position, with the difference being significant (*p* < 0.05) and shorter by 0.02 s. These findings suggest that a narrow base position on the start block produces better exit values and performance parameters compared to a wide one. In terms of performance over the 5 m distance, in our study, the difference between the wide and narrow base foot stances was 0.08 s and in favor of the narrow stance. Such a significant difference (*p* < 0.05) may determine the ranking, especially in sprinters’ events. For example, at the 2023 World Championships in Fukuoka, the difference in the 50 m freestyle between the second and third place was 0.01 s, third and fourth place 0.12 s, fourth and fifth place 0.02 s, fifth and sixth place 0.01 s, etc. (WA, 2023) [34]. The results point to the fact that in such a discipline, milliseconds decide the ranking and, therefore, it is necessary to perform an analysis of the start and its phases for each swimmer, with a subsequent transfer to the training process to make the performance in the start more efficient. If coaches cannot perform an analysis, they should be guided by available information from the literature that they can use in coaching the start to potentially improve performance. We are aware that one of the limitations of the study is the sample size. On the other hand, performance swimmers who have similar performances in the 50 m freestyle were selected for the sample, and likewise, this sample was confirmed to be homogeneous in the kinematic analysis of performance in the 5 m start.

## 5. Conclusions

Swimmers were found to have significant differences (*p* < 0.05) between foot widths in the baseline position when taking off from the starting block at the 5 m distance in favor of the narrow baseline position. Differences in foot position in the base position were also reflected in the kinematic parameters in the phases after take-off from the start block, in block time in the phase on the start block, in flight time and distance in the flight phase with time at 2 m, and in all parameters observed in the underwater phase. The differences between the final time at the start block (0.02 s) and the time at 2 m (0.05 s) were approximately twofold and fourfold with the time at 5 m distance (0.08 s) when the swimmer was underwater. Based on our findings, we recommend marking the center of the start block on the OSB or OSB platform for the competitors, as well as the center of the backrest, for better orientation and assuming the correct basic foot position on the start block.

## Figures and Tables

**Figure 1 sports-12-00042-f001:**
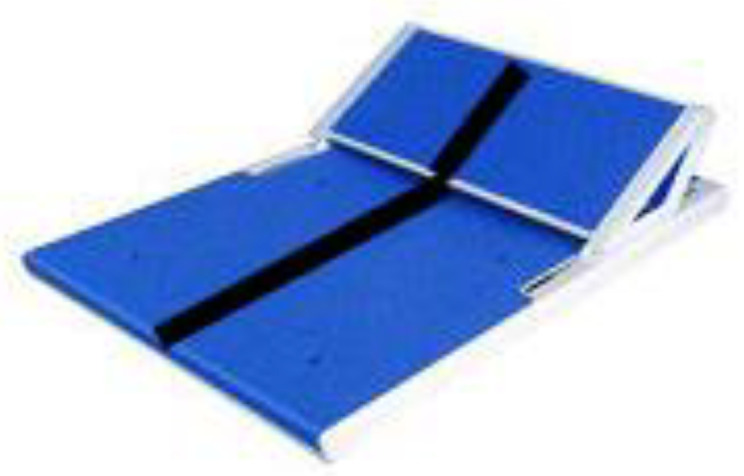
Starting platform OSB12.

**Figure 2 sports-12-00042-f002:**
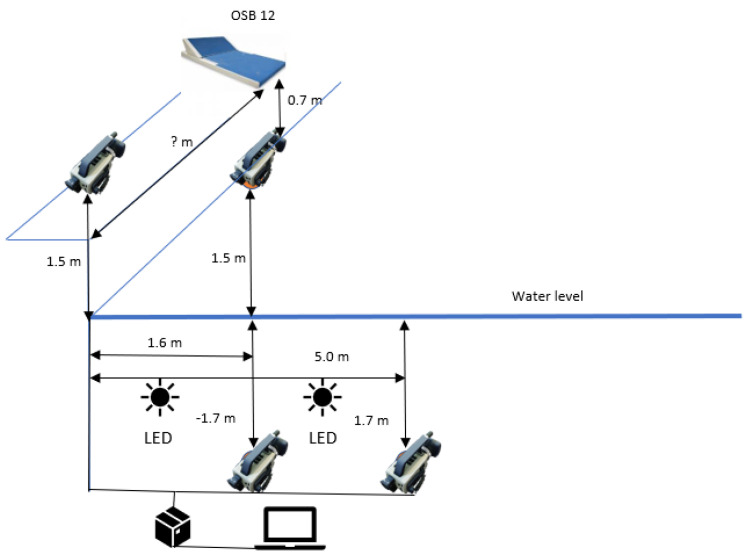
Swimpro camera set–up.

**Table 1 sports-12-00042-t001:** Comparison of block phase parameters between the narrow and wide positions.

Variables	Narrow		Wide		ES	*t*	Prob	Prob	Prob
	*M*	*SD*	*M*	*SD*	Cohen’s *d*		(T < *t*)	(|T| > |*t*|)	(T > *t*)
FKA (°)	133.19	3.08	132.67	3.26	0.17	0.64	0.74	0.53	0.26
FAA (°)	127.33	4.52	126.84	4.58	0.11	0.42	0.66	0.68	0.34
RKA (°)	82.45	2.40	82.06	2.35	0.17	0.64	0.74	0.53	0.26
RAA (°)	100.03	3.31	99.41	3.44	0.18	0.71	0.76	0.48	0.24
HA (°)	43.70	2.02	43.4	1.63	−0.02	−0.07	0.47	0.94	0.53
BT (s)	0.77	0.03	0.79	0.02	−0.72	−2.78	0.001 *	0.001 *	1.00

Note: FKA front knee angle, FAA front ankle angle, RKA rear knee angle, RAA rear ankle angle, HA hip angle, BT block time, * *p* < 0.05.

**Table 2 sports-12-00042-t002:** Comparison of flight phase parameters between the narrow and wide positions.

Variables	Narrow		Wide		ES	*t*	Prob	Prob	Prob
	*M*	*SD*	*M*	*SD*	Cohen’s *d*		(T < *t*)	(|T| > |*t*|)	(T > *t*)
TA (°)	39.77	2.39	39.67	2.67	0.04	0.15	0.56	0.88	0.44
T2 (s)	0.94	0.02	0.99	0.02	−2.11	−8.18	0.001 *	0.001 *	1.00
T2 (m/s)	2.13	0.06	2.02	0.5	2.16	8.37	1.00	0.001 *	0.001 *
EA (°)	36.80	0.85	36.69	1.35	0.09	0.34	0.63	0.73	0.37
FT (s)	0.37	0.01	0.41	0.01	−3.40	−13.18	0.001 *	0.001 *	1.00
FD (m)	2.69	0.07	2.61	0.07	1.12	4.35	1.00	0.001 *	0.001 *

Note: TA take-off angle, T2 time to 2 m, T2 velocity to 2 m, EA entry angle, FT flight time, FD flight distance, * *p* < 0.05.

**Table 3 sports-12-00042-t003:** Comparison of underwater phase parameters between the narrow and wide positions.

Variables	Narrow		Wide		ES	*t*	Prob	Prob	Prob
	*M*	*SD*	*M*	*SD*	Cohen´s *d*		(T < *t*)	(|T| > |*t*|)	(T > *t*)
GT (s)	0.52	0.02	0.55	0.02	−2.11	−8.16	0.001 *	0.001 *	1.00
GD (m)	2.31	0.07	2.39	0.07	−1.13	−4.36	0.001 *	0.001 *	1.00
MaxH (m)	−0.89	0.02	−0.92	0.02	1.4	5.44	1.00	0.001 *	0.001 *
T5 (s)	1.67	0.03	1.75	0.03	−3.41	−13.22	0.001 *	0.001 *	1.00
T5 (m/s)	2.99	0.06	2.85	0.05	3.40	13.19	1.00	0.001 *	0.001 *

Note: GT glide time, GD glide distance, MaxH maximal depth, T5 time to 5 m, T5 velocity to 5 m, * *p* < 0.05.

## Data Availability

The data used in this study will be made available from the author upon reasonable request.

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
