# Peer review of "Foot Placement in the Basic Position on the Start Block OSB12 of Young Competitive Swimmers"

_sports, 2024, doi:10.3390/sports12020042_

Round 1

Reviewer 1 Report

Comments and Suggestions for Authors

I want to thank the journal's editorial board for the opportunity to review this interesting paper.

Firstly, in my opinion, the paper is structured correctly; the structure is impeccable. This is appreciated as it greatly facilitates analysis and reading. Moreover, this study contributes to enhancing performance in swimming competition. Marks are increasingly difficult to surpass, and therefore, the margin for improvement is decreasing, achievable only through technical enhancements and parameter optimization based on scientific evidence, as presented in this paper.

Lines 84-93. In my opinion, the sample size is very small, but the difficulty in obtaining a larger sample in these categories should be considered. I suggest the authors mention this limitation in the study's limitations, specifically in the "Discussion" section. Figure 1 is unnecessary as the model is well-known in the field of swimming. It's important to note that specialists in the subject matter read these types of papers. In the "Materials and Methods" section, it is necessary to add a separate subsection on statistical analysis, detailing the tests used along with their interpretation scales. While the authors include this information, I believe it should be presented as an independent section to enhance the paper's readability.

I believe there might be an error in line 84. The authors indicate (1.86±0.07 cm), but I think it should be in meters, not centimeters. My recommendation is for the authors to convert the units to centimeters.

Line 179, the p-value can never be 0. Sometimes, the software might display 0, but this is incorrect. In such cases, it should be indicated as p-value<0.001.

Tables 1, 2, and 3: Only the t-test probability of Pr(|T| > |t|) should be reported; the remaining columns with p-values should be removed as it doesn't make sense to indicate those values. Additionally, p-values of 0.00 should be expressed as <0.001. Moreover, if the p-value is indicated, asterisks are unnecessary. Asterisks should be used to denote differences in means, not p-values.

In the discussion, the study's limitations should be highlighted, including the sample size.

I recommend that the authors "get straight to the point" in the conclusions. As they are currently written, they are a mixture of results, discussion, and conclusions. For instance, the phrase (lines 326-317) "Although these differences seem small, in sprint events 0.01 s is decisive" is more suited for the discussion section, not the conclusions. Therefore, they should summarize the main conclusions and the important aspects. If readers seek further information, they can find it in the "Results" section. Additionally, phrases such as "...to have significant differences (p<0.05)..." are unnecessary in the conclusions as the significance of p<0.05 is widely known in research. Reminding it in the conclusions doesn't add value.

Author Response

Reviewer 1

  1. Note

Lines 84-93. In my opinion, the sample size is very small, but the difficulty in obtaining a larger sample in these categories should be considered. I suggest the authors mention this limitation in the study's limitations, specifically in the "Discussion" section. Figure 1 is unnecessary as the model is well-known in the field of swimming. It's important to note that specialists in the subject matter read these types of papers. In the "Materials and Methods" section, it is necessary to add a separate subsection on statistical analysis, detailing the tests used along with their interpretation scales. While the authors include this information, I believe it should be presented as an independent section to enhance the paper's readability.

Answer: edited – line 308-312

  1. Note

I believe there might be an error in line 84. The authors indicate (1.86±0.07 cm), but I think it should be in meters, not centimeters. My recommendation is for the authors to convert the units to centimeters.

Answer: edited – line 84

  1. Note

Line 179, the p-value can never be 0. Sometimes, the software might display 0, but this is incorrect. In such cases, it should be indicated as p-value<0.001.

Tables 1, 2, and 3: Only the t-test probability of Pr(|T| > |t|) should be reported; the remaining columns with p-values should be removed as it doesn't make sense to indicate those values. Additionally, p-values of 0.00 should be expressed as <0.001. Moreover, if the p-value is indicated, asterisks are unnecessary. Asterisks should be used to denote differences in means, not p-values

Answer: edited – line 179 and table 1

  1. Note

In the discussion, the study's limitations should be highlighted, including the sample size.

Answer: edited – line 308-312

  1. Note

I recommend that the authors "get straight to the point" in the conclusions. As they are currently written, they are a mixture of results, discussion, and conclusions. For instance, the phrase (lines 326-317) "Although these differences seem small, in sprint events 0.01 s is decisive" is more suited for the discussion section, not the conclusions. Therefore, they should summarize the main conclusions and the important aspects. If readers seek further information, they can find it in the "Results" section. Additionally, phrases such as "...to have significant differences (p<0.05)..." are unnecessary in the conclusions as the significance of p<0.05 is widely known in research. Reminding it in the conclusions doesn't add value.

Answer: edited – line 313-323

Reviewer 2 Report

Comments and Suggestions for Authors

The manuscript is engaging and well-written; however, I would like to make suggestions to the authors to improve the manuscript.

The abstract is well-written and clear.

Introduction.

The purpose of the study should be rewritten in a clearer and more complete manner

Methods 

-   Maybe I missed it, but it would be useful for the authors to better describe how the sample was recruited.

-   Has the sample power been calculated?

-   Are the athletes all in the same category/level? This is unclear.

-   What is the validity and reliability of the device used to measure knee ROM?

-   The statistics subparagraph should be rewritten and divided from the protocol, furthermore, there is no information on the software used.

Results: 

-   A graph showing the recorded performances would give clarity to the reader.

-   I would suggest the authors do a Pearson correlation between the parameters. It would be interesting to evaluate whether there was any relationship between parameters and performance. (block phase – FKA front knee angle (°), FAA front ankle angle (°), RKA rear knee 1angle (°), RAA rear ankle angle (°), HA hip angle (°), BT block time (s); flight phase …….)

Discussion 

-   The authors examine the start techniques well; however, the performance in the start phase is an experience of explosive force. Some studies have shown that joints such as the knee can influence force transmission and therefore flight performance. Giustino et al studied the relationship of knee position on jumping performance. This is certainly valid also in the block phase of swimmers. I suggest strengthening the discussion by expanding on the reasons beyond the position of the feet. (Giustino V, et al. Effects of a Postural Exercise Program on Vertical Jump Height in Young Female Volleyball Players with Knee Valgus. Int J Environ Res Public Health. 2022 Mar 26;19(7):3953. doi: 10.3390/ijerph19073953).

-   I did not find limitations in the study. should be included.

-   The discussion is generally well written, but some statements are in my opinion speculative data unless supported by more comprehensive statistics. I would also like to evaluate whether there are correlations between the recorded parameters. I would suggest rephrasing to say "This pilot study raises the possibility that..."

Author Response

  1. Note

-   Has the sample power been calculated?

Answer - deliberate selection on the basis of performance. The average FINA score in the 50 m freestyle short distance pool in the 1 month prior to the study was 630 points

-   Are the athletes all in the same category/level? This is unclear.

Answer – competitive swimmers

-   What is the validity and reliability of the device used to measure knee ROM?

Answer- A high level of validity and reliability was reported in previous research by the authors mentioned in the 2D analysis methodology: Norris & Olson (2011) / Pearson r ≥ 0.95 / ICC ≥ 0.91

-   The statistics subparagraph should be rewritten and divided from the protocol, furthermore, there is no information on the software used.

Answer – edited – line 171

  1. Note

-   I would suggest the authors do a Pearson correlation between the parameters. It would be interesting to evaluate whether there was any relationship between parameters and performance. (block phase – FKA front knee angle (°), FAA front ankle angle (°), RKA rear knee 1angle (°), RAA rear ankle angle (°), HA hip angle (°), BT block time (s); flight phase …….)

Answer - correlation analysis was already done in the previous article - Matúš, I.; Ružbarský, P.; Vadašová, B. Key Parameters Affecting Kick Start Performance in Competitive Swimming. Int. J. Environ. Res. Public Health 2021, 18, 11909, doi:10.3390/ijerph182211909.

  1. Note

Discussion

-   The authors examine the start techniques well; however, the performance in the start phase is an experience of explosive force. Some studies have shown that joints such as the knee can influence force transmission and therefore flight performance. Giustino et al studied the relationship of knee position on jumping performance. This is certainly valid also in the block phase of swimmers. I suggest strengthening the discussion by expanding on the reasons beyond the position of the feet. (Giustino V, et al. Effects of a Postural Exercise Program on Vertical Jump Height in Young Female Volleyball Players with Knee Valgus. Int J Environ Res Public Health. 2022 Mar 26;19(7):3953. doi: 10.3390/ijerph19073953).

Answer – edited – line 272-273

  1. Note

-   I did not find limitations in the study. should be included.

Answer – edited – line 308-311

  1. Note

The discussion is generally well written, but some statements are in my opinion speculative data unless supported by more comprehensive statistics. I would also like to evaluate whether there are correlations between the recorded parameters. I would suggest rephrasing to say "This pilot study raises the possibility that..."

Answer – edited – 313-323

Round 2

Reviewer 2 Report

Comments and Suggestions for Authors

I thank the authors for the work done. I have no other suggestions to make